# Development of a Kit for Rapid Immunochromatographic Detection of Sacbrood Virus Infecting *Apis cerana* (AcSBV) Based on Polyclonal and Monoclonal Antibodies Raised against Recombinant VP1 and VP2 Expressed in *Escherichia coli*

**DOI:** 10.3390/v13122439

**Published:** 2021-12-04

**Authors:** Song Hee Lee, Tae-Kyun Oh, Sung Oh, Seongdae Kim, Han Byul Noh, Nagarajan Vinod, Ji Yoon Lee, Eun Sun Moon, Chang Won Choi

**Affiliations:** 1Department of Biology & Medicinal Science, Pai Chai University, Daejeon 35345, Korea; thdgml6245@naver.com (S.H.L.); ohsung85@gmail.com (S.O.); khboy111@pcu.ac.kr (S.K.); creator1018@pcu.ac.kr (H.B.N.); biovinz@gmail.com (N.V.); spunji970@naver.com (J.Y.L.); ansdmstjs124@naver.com (E.S.M.); 2GeNet Bio Company, Daejeon 305-500, Korea; tkoh@genetbio.co.kr

**Keywords:** sacbrood virus infecting *Apis cerana* (AcSBV), recombinant VP1 (rVP1) and VP2 (rVP2) proteins, polyclonal antibodies (pAb-rVP1 and pAb-rVP2), monoclonal antibodies (mAb-rVP1 and mAb-rVP2), immunochromatography (IC) strip assay

## Abstract

A Korean isolate of the sacbrood virus infecting *Apis cerana* (AcSBV-Kor) is the most destructive honeybee virus, causing serious economic damage losses in Korean apiculture. To address this, here, we attempted to develop an assay for the rapid detection of AcSBV-Kor based on immunochromatographic detection of constituent viral proteins. Genes encoding VP1 and VP2 proteins of AcSBV-Kor were cloned into an expression vector (pET-28a) and expressed in *Escherichia coli* BL21(DE3). During purification, recombinant VP1 (rVP1) and VP2 (rVP2) proteins were found in the insoluble fraction, with a molecular size of 26.7 and 24.9 kDa, respectively. BALB/c mice immunized with the purified rVP1 and rVP2 produced polyclonal antibodies (pAbs) such as pAb-rVP1 and pAb-rVP2. Western blot analysis showed that pAb-rVP1 strongly reacted with the homologous rVP1 but weakly reacted with heterologous rVP2. However, pAb-rVP2 strongly reacted not only with the homologous rVP2 but also with the heterologous rVP1. Spleen cells of the immunized mice fused with SP2/0-Ag14 myeloma cells produced monoclonal antibodies (mAbs) such as mAb-rVP1-1 and mAb-rVP2-13. Western blot analysis indicated that pAb-rVP1, pAb-rVP2, mAb-rVP1-1, and mAb-rVP2-13 reacted with AcSBV-infected honeybees and larvae as well as the corresponding recombinant proteins. These antibodies were then used in the development of a rapid immunochromatography (IC) strip assay kit with colloidal gold coupled to pAb-rVP1 and pAb-rVP2 at the conjugate pad and mAb-rVP1-1 and mAb-rVP2-13 at the test line. One antibody pair, pAb-rVP1/mAb-VP1-1, showed positive reactivity as low as 1.38 × 10^3^ copies, while the other pair, pAb-rVP2/mAb-VP2-13, showed positive reactivity as low as 1.38 × 10^4^ copies. Therefore, the antibody pair pAb-rVP1/mAb-VP1-1 was selected as a final candidate for validation. To validate the detection of AcSBV, the IC strip tests were conducted with 50 positive and 50 negative samples and compared with real-time PCR tests. The results confirm that the developed IC assay is a sufficiently sensitive and specific detection method for user-friendly and rapid detection of AcSBV.

## 1. Introduction

Sacbrood virus (SBV) is one of the most destructive honeybee viruses and it causes economic losses in Asian apiculture [1,2,3,4,5,6,7]. Chinese sacbrood virus (CSBV) reemerged in Liaoning Province of China in 2008, devastating the local apiculture [8]. The threat of mass extinction of the *A. cerana* population is an ongoing problem in Korea because the lethal AcSBV-Kor has caused more than 90% mortality since its first occurrence in 2008 [9,10,11,12]. SBV is a single-stranded, positive-sense RNA (~8.8 kb in size) virus that belongs to the family *Iflaviridae* [13]. SBV isolates infect *Apis cerana* (AcSBV) and *A*. *mellifera* (AmSBV), each of which represents different serotypes [14,15,16]. In a previous cross-infection study, AmSBV and AcSBV were shown to be pathogenic only in *A*. *mellifera* larvae and *A*. *cerana* larvae, respectively [17]. However, a recent study showed that AcSBV could infect *A*. *mellifera* with low pathogenicity [15], and phylogenetic tree analyses supported cross-infection between AmSBV and AcSBV [5,7,16,18,19].

Four structural proteins in the order VP2–VP4–VP3–VP1 are located at the SBV capsid [19], although the presence of the VP4 protein has yet not been demonstrated [13]. The capsid proteins VP1, VP2, and VP3 are processed from one polyprotein precursor (P1 protein) to form a protomer and organized with pseudo-T3 icosahedral symmetry. The VP1 subunits form pentamers positioned around a 5-fold axis, whereas the VP2 and VP3 subunits form hetero-hexamers distributed at the icosahedral 3-fold axis [13,20]. Recently, a Korean isolate of the AcSBV (AcSBV-Kor) was detected using a polyclonal antibody (pAb) raised against the recombinant protein VP3 (rVP3, 26.18 kDa) expressed in *Escherichia coli* [21]. In addition, AcSBV was detected from AcSBV-infected honeybees using a monoclonal antibody (mAb) raised against the rVP2 protein expressed in *E*. *coli* [22].

To date, reverse transcriptase polymerase chain reaction (RT-PCR), reverse transcription loop-mediated isothermal amplification (RT-LAMP), and real-time RT-PCR are the most conventional methods for the detection and quantification of the viral genome of bee-infecting viruses [1,9,23,24,25,26,27,28,29]. Recently, a next-generation sequencing method was used to discover additional bee-infecting viruses [30]. Nevertheless, these methods cannot rapidly diagnose AcSBV infection in larva and honeybee samples. At present, no user-friendly antigen–antibody-based test kits are available. Therefore, it is necessary to find a target antigen to produce an antibody for the antigen point-of-care testing (POCT) and rapid AcSBV diagnosis.

Over the last few decades, many immunochromatography (IC) assays have been developed to diagnose human or animal infectious diseases [31,32,33,34,35,36,37]) and to detect specific markers in the medical [38,39,40], agricultural, and food industries [41,42,43]. An IC strip is composed of four components, a sample pad, a conjugate pad, a nitrocellulose membrane, and an absorbent pad [44]. In general, antibody pairs have been used for the IC strip using either a single monoclonal antibody (mAb) [45] or two different mAbs as detector and capture antibodies [34,46,47]. The combination of a mAb and a polyclonal antibody (pAb) as detector and to capture antibodies provided an alternative platform to the sandwich format of IC test [32] and vice versa [41].

In this study, genes encoding VP1 and VP2 proteins of (AcSBV-Kor were cloned into an expression vector (pET-28a) and expressed in *Escherichia coli* BL21(DE3) for production of the recombinant VP1 (rVP1) and VP2 (rVP2) proteins. BALB/c mice were immunized with the purified rVP1 and rVP2 to produce pAbs and mAbs, respectively. Ultimately, we developed an IC assay using the antibody pair (pAb as detector antibody and mAb as capture antibody) for the rapid, sensitive, and specific detection of AcSBV in larvae and honeybees collected from a beekeeping farm.

## 2. Materials and Methods

### 2.1. Synthesis of VP1 and VP2 DNA and Construction of Gene Expression Vector

Fragments containing *VP1* (579 bp) or *VP2* (528 bp) genes were amplified from the pBHA vector containing a full length of the capsid gene of AcSBV-Kor [21]. Primers used in this work were designed based on the nucleotide sequences of AcSBV-Kor (GenBank accession no. HQ322114): the *VP1* gene (forward: 5′-TCGCGGATCCGAATTCGACATTTTGCGTAGACCAGTGTTGT-3′ and reverse: 5′- CCGCAAGCTTGTCGACTGGCCCATAAAAGTTAGACAC CTC-3′) and the *VP2* gene (forward: 5′-TCGCGGATCCGAATTCT GGATGCCTATAAATTCAATTAAGGTACA-3′ and reverse: 5′-CCGCA AGCTTGTCGACGACTTTGTACGACATTCCCGCAAAT-3′). *Eco*RI and *Sal*I sites (underlined) were engineered into each primer. PCR reaction (50 μL) contained 5× PCR buffer (10 μL), 10 mM dNTPs (5 μL), 10 pmol of each primer, 1 ng plasmid DNA, and 0.4 μL (1 U) SuPrime HF DNA Polymerase (GeNet Bio, Daejeon, Korea). The reaction was carried out under the following conditions: 1 cycle of 98 °C for 2 min; 35 cycles of 98 °C for 30 s, 58 °C for 30 s, and 68 °C for 2 min; and an additional 1 cycle of 68 °C for 5 min. Each amplified DNA was double-digested with *Eco*RI/*Sal*I and ligated into Novagen^®^ pET-28a (+) (MilliporeSigma, Burlington, MS, USA) to yield the recombinant vector. The resultant vectors (pET28a-VP1 and pET28a-VP2) were transformed into *E*. *coli* BL21(DE3) and inoculated into Luria–Bertani (LB) agar medium containing kanamycin (50 μg/mL) and grown overnight at 37 °C.

### 2.2. Inducible Expression and Purification of Recombinant Protein and SDS-PAGE

A selected single colony was cultured at 37 °C in LB liquid medium supplemented with kanamycin (50 μg/mL) until the OD_600 nm_ reached 0.6 and was then induced by adding isopropyl β-D-1-thiogalactopyranoside (IPTG) at a final concentration of 1.0 mM at 37 °C for 4 h. Bacterial cells were harvested by centrifugation and further subjected to sonication in lysis buffer (10 mM imidazole, 300 mM NaCl, 50 mM NaH_2_PO_4_, pH 8.0). Two recombinant proteins (rVP1 and rVP2) were purified from *E*. *coli* cells using a Ni-NTA resin column under denaturing conditions according to the manufacturer’s manual (Qiagen, Hilden, Germany), and recombinant proteins were analyzed by sodium dodecyl sulfate-polyacrylamide gel electrophoresis (SDS-PAGE) as described previously [21].

### 2.3. Immunization and Production of Polyclonal Antibody

All animal procedures were approved by the Ethics Committee of Pai Chai University. The purified rVP1 and rVP2 proteins were dialyzed twice against 1× phosphate-buffered saline (PBS) for 1.5 h at 4 °C and used as immunogens. The polyclonal antiserum was prepared by injecting three BALB/c mice (male, six-week-old) with each recombinant protein. The injection schedule, method, and materials were the same as described previously [21]. Three immunizations were given to mice in 2-week intervals, and each antiserum was recovered from the mouse’s blood one week after every injection. The purified rVP1 and rVP2 (2000, 1000, 500, 250, 125, and 62.5 ng/20 μL) were separated by 10% SDS-PAGE and tested with each antiserum by Western blot analysis as described below. Furthermore, the immunoglobulin G (IgG) was purified using Montage^™^ Antibody Purification Kits with Prosep-G media according to the manufacturer’s protocol (MilliporeSigma) and adjusted to 1 mg/mL concentration.

### 2.4. ELISA, SDS-PAGE, and Western Blot Analysis

To test IgG purified from each antiserum (6-week collection), an aliquot (50 μL) of antigen (31.25, 62.5, 125, 250, and 500 ng/mL of rVP1 and rVP2 in coating buffer) was added to a microtiter plate. The plate was blocked with PBS-T (PBS containing 0.05% Tween-20) containing 1% bovine serum albumin (BSA)at room temperature for 2 h. After washing the plate with PBS-T, pAb-rVP1 and pAb-rVP2 (1:1000, *v*/*v* in blocking buffer) were added and incubated at room temperature for 2 h. After adding alkaline phosphatase-conjugated goat anti-mouse IgG (1:5000, *v*/*v* in blocking buffer, MilliporeSigma), the plate was incubated at room temperature for 2 h. After washing with PBS-T, p-nitrophenyl phosphate ( MilliporeSigma) was added and further incubated at room temperature for 2 h. The absorbance at 450 nm was measured using a microplate reader (Bio-Rad, Hercules, CA, USA). A sample was considered positive if the mean absorbance value of the six replicates was more than double that of the PBS buffer control. SDS-PAGE gels (10%) separated the purified rVP1 and rVP2 proteins, as well as crude protein extracts from healthy honeybees, AvSBV-infected honeybees, and larvae (Samcheok, Gangwon-do, Korea). The unstained gel was blotted onto a polyvinylidene fluoride (PVDF) membrane (MilliporeSigma) using wet electro-transfer apparatus (Bio-Rad). The membrane was blocked with skim milk (5%) in Tris-buffered saline (TBS; 137 mM sodium chloride, 20 mM Tris, pH 7.4) for 1 h at 37 °C. The membrane was washed thrice in TBS-T (0.1% Tween-20 in TBS) and incubated at 4 °C overnight with a primary antibody (1:1000 dilution of pAb-rVP1 and pAb-rVP2, *v*/*v*). After washing thrice with TBS-T, the membrane was incubated at 37 °C for 1 h with a secondary antibody (1:10,000 dilution of alkaline phosphatase-conjugated goat anti-mouse antibody, *v*/*v*). Then, nitro blue tetrazolium (100 μL) was added to alkaline phosphatase buffer (15 mL) before mixing with 5-bromo-4-chloro-3-indolylphosphate (50 μL) to stain the membrane.

### 2.5. Preparation of Splenocyte and Hybridoma Production and Selection

Hybridomas were generated using the ClonaCell-HY Hybridoma Cloning Kit according to the manufacturer’s instructions (Stemcell Technologies Inc., Vancouver, BC, Canada). The spleen was disaggregated into a single-cell suspension in Medium B and the cell suspension was centrifuged at 300× *g* for 10 min at room temperature (a total of 3 cycles). After resuspension of the cells in Medium B (25 mL), and the volume of cell suspension was calculated (1 × 10^8^ cells). The myeloma cells (SP2/0-Ag14, CRL-1581) in Medium A were incubated for 4 days (37 °C, 5% CO_2_), and the cell suspension was centrifuged at 300× *g* for 10 min. After adding additional Medium B to the supernatants, the cell suspension was centrifuged at 300× *g* for 10 min (a total of 3 cycles). After resuspension of the cells in Medium B (25 mL), the volume of cell suspension was calculated (2 × 10^7^ cells). Myeloma cells were fused with spleen cells using ClonaCell^™^-HY PEG (0.5 mL), and Medium B (5 mL) was added to the fusion mixture. After dropwise adding Medium C (5 mL), the mixture was incubated for 24 h (37 °C, 5% CO_2_). The fused cell suspension was mixed with Medium D vigorously, transferred into a 50 mL conical tube, and centrifuged at 400× *g* for 10 min. The cells were resuspended in Medium C (made up to 3 mL), transferred into a bottle containing Medium D (10 mL), gently inverted several times, and incubated at room temperature for 15 min. The cell suspension in Medium D (9.5 mL) was slowly added to Petri dishes using a 12 mL syringe with a 16-gauge blunt-end needle and incubated for 10–14 days. Each clone was pipetted into an individual well of a 96-well tissue culture plate containing Medium E (200 µL), and the plates were incubated for 4 days. To achieve monoclonality, hybridoma cell suspension (100 cells/1 mL) was mixed with Medium D (9 mL). The mixture was plated in a Petri dish and incubated for 14 days. Twenty-four clones were incubated in 96-well microplates containing Medium E (200 μL/well), and supernatants from the microplates were screened by Western blotting using the purified rVP1 and rVP2.

### 2.6. Generation of Ascities Fluids

After screening by Western blotting, the positive hybridoma cells (4 × 10^6^) were suspended in Medium E and injected into the peritoneal cavity of a Freund’s incomplete adjuvant (0.5 mL)-primed mouse using a 26-gauge needle. After 10 days, the mouse developed a large quantity of ascites fluid, and the abdominal cavity was opened by cutting after sacrificing the mouse. Ascites fluid was drawn using a 10 mL syringe fitted with an 18-gauge needle, aseptically collected from anesthetized mice by abdominal paracentesis using an 18-gauge needle, and then transferred into sterile centrifuge tubes. The fluid was centrifuged at 200× *g* for 10 min at 4 °C. The supernatant fluid was collected and frozen at −70 °C until further use.

### 2.7. Purification of mAbs, SDS-PAGE, and Western Blot Analysis

Immunoglobulin G (IgG) was purified using a Proteus Antibody Purification Kit Protein A (Bio-Rad) according to the manufacturer’s protocol (MilliporeSigma). The concentrations of mAb-rVP1 and mAb-rVP2 were adjusted to 1 mg/mL, mixed with glycerol (50%, *v*/*v*), and stored at −20 °C until use. Crude protein extracts from the healthy honeybee, AvSBV-infected honeybee, and larva samples were included. Diluted mAb-rVP1 and mAb-rVP2 (1:1000, *v*/*v*) were used as primary antibodies, and diluted goat anti-mouse phosphatase conjugate (1:10,000, *v*/*v*, MilliporeSigma) was used as a secondary antibody.

### 2.8. Preparation of Colloidal Gold and Colloidal Gold−pAb Conjugate

Firstly, 0.01% HAuCl_4_ in double-distilled water was boiled with rapid stirring, and 1% trisodium citrate (2 mL) was added to the solution. The colloidal gold solution was boiled for an additional 10 min and continuously stirred until it had cooled. Secondly, the colloidal gold solution was adjusted to pH 8.2 using 1% potassium carbonate. Thirdly, each pAb (300 μL of 1 mg/mL) was mixed with 20 mL of a colloidal gold solution by vigorous stirring for 30 min. The mixture was added to 2.5 mL of 10% bovine serum albumin and stirred for an additional 30 min. The mixture was centrifuged at 6000× *g* for 45 min at 4 °C, then the resulting conjugate pellet was resuspended and washed twice with 2 mM borax buffer (pH 9.0) containing 0.1% PEG 20000, followed by resuspension in 1 mL of the same buffer.

### 2.9. Preparation of IC Strip Device and Procedure for the Test

The sample pads (Millipore cellulose fiber: MilliporeSigma) and the conjugate pads (Millipore glass-fiber membrane: MilliporeSigma) were treated with 20 mM phosphate buffer containing 2% BSA, 2.5% sucrose, 1% Tween 20, 0.3% polyvinylpyrrolidone K30, and 0.02% sodium azide (pH 7.4) and dried for 1 h at 37 °C. The mAb-rVP1, mAb-rVP2 (1 mg/mL), and the goat anti-mouse antibody (1 mg/mL) in 1× PBS were dispensed at the test line and the control line on the Millipore nitrocellulose membrane. The pAb-rVP1 and pAb-rVP2 colloidal gold conjugates were applied to the treated conjugate pad at a rate of 10 μL/cm (about 1.5 g/cm) and then completely lyophilized. The absorption pad, nitrocellulose membrane, pretreated conjugate pad, and sample pad were assembled as a test strip and attached to a plastic scale board. A complete set for the IC strip was manufactured by GeNet Bio Co (Daejeon, Korea). Samples were prepared from larvae (0.3~0.5 cm) in Eppendorf tubes containing 1× PBS using a disposable plastic homogenizer. The resulting extract was centrifuged at 8000 rpm for 30 s, and the supernatant was used as a liquid sample. During the assay, a single drop (50 μL/drop) of the liquid sample and two drops of diluent buffer (1× PBS) were added to the sample pad, and it rapidly diffused into the conjugate pad. If a sample contains the target antigens (VP1 and VP2), it reacts with the colloidal gold-conjugated pAb-rVP1 and pAb-rVP2 to form an antigen colloidal gold–pAb complex. The complex migrates along the nitrocellulose membrane chromatographically and reacts with immobilized mAb-rVP1 and mAb-rVP2 on the test line to form a colored band. The excess conjugate, or free conjugate if the sample does not contain the target antigen, migrates along the membrane to the control line, where it interacts with immobilized goat anti-mouse antibody to form a colored band. Therefore, two bands will be displayed for a positive sample, one at the test line and one at the control line, while a negative sample will show only one band at the control line within 10 min.

### 2.10. Real-Time RT-PCR

Total RNA was isolated from larvae using RNAiso Plus according to the manufacturer’s instructions (Takara Bio, Shiga, Japan). The RT reaction was performed in a reaction mixture (20 μL) containing 1 µL of total RNA (100 ng), 10 µL of 2× SuPrimeScript RT premix, and 1 µL of 20 pmol of reverse primer (GeNet Bio, Daejeon, Korea). The thermal cycler was programmed for 1 RT cycle at 50 °C for 60 min and 70 °C for 10 min. PCR amplification was performed in 96-well plates by using an Applied Biosystem 7500 Real-Time PCR Instrument System. The reaction mixture (20 µL) contained 5 µL of template DNAs, 10 µL of 2× SYBR Green I-based Prime Q-Master Mix (GeNet Bio, Daejeon, Korea), 5 µL of 4× oligo mixture, and primer pairs at 10 pmol. A primer set was designed to cover a part of the *VP1* gene (forward: 5′-CATGGAGAGACAAA GGCGATAC-3′ and reverse: 5′-GCTTCTACCCACACAGACATATT-3′) and the *VP2* gene (forward: 5′-TGCGCGCCCA ATACTATAC-3′ and reverse: 5′-CTTTCCCGCACTGAAACTTATTAC-3′). The reaction was performed under the following conditions: one cycle of initial denaturation at 95 °C for 10 min, 40 cycles of denaturation at 95 °C for 10 s, and annealing extension at 60 °C for 30 s. The experiments were analyzed with auto-baseline and manual thresholds chosen from the exponential phase of the PCR amplification. After the data analysis, the cycle threshold (Ct) number and DeltaRn (dRn) were used for statistical analyses. Data were analyzed using the Applied Biosystems^™^ High Resolution Melting (HRM) Software.

For the standard of DNA copy number, we used pET28a-VP1 containing a 579 bp insert and pET28a-VP2 containing a 528 bp insert. A ten-fold serial dilution series of the pET28a-VP1 and pET28a-VP2 ranging from 1 × 10^3^ to 1 × 10^6^ copies/μL, was used to construct the standard curves for both VP1 andVP2. The concentration of the plasmid was measured using a fluorometer and the corresponding copy number was calculated using the following equation: DNA (copy) = 6.02 × 10^23^ (copy/mol) × DNA amount (g)/DNA length (bp) × 660 (g/mol/bp) [48]. The plot of a standard curve of Ct values against the logarithmic dilutions produced a regression line (y = −3.8356x + 42.7930, R^2^ = 0.9965) for the VP1 and a regression line (y = −3.7314x + 43.5159, R^2^ = 0.9933) for the VP2. The data was shown as Appendix A.

### 2.11. Detection Limit (Sensitivity) of the IC strips and Validation (Specificity) of the IC Strips Using Field Samples

A total of 100 larva samples were collected from two beekeeping farms in the Sejong (Chungcheongnam-do) and Samcheok (Gangwon-do) areas of Korea and tested by real-time PCR for the presence of AcSBV. The limit of antibody pairs (pAb-rVP1/mAb-VP1-1 and pAb-rVP2/mAb-VP2-13) embedded in a membrane-based strip in detecting AcSBV was determined by using serial dilutions of real-time PCR positive reference samples (3.02 × 10^6^, 4.33 × 10^5^, 1.38 × 10^4^, 1.38 × 10^3^ copies). In addition, real-time PCR-positive (50 infected) and -negative (50 healthy) samples were tested with IC strips for the validation of AcSBV diagnosis. Each diluted sample was tested in triplicate using different IC strips.

## 3. Results

### 3.1. Expression and Purification of the Recombinant Protein VP1 (rVP1) and VP2 (rVP2)

VP1 and VP2 are located at the C-terminus and N-terminus of AcSBV capsid protein (Figure 1A). Both rVP1 and rVP2 were expressed as inclusion bodies (Figure 1B), suggesting that these recombinant proteins were not soluble in *E. coli*. Prominent bands at the expected molecular weights of 26.73 kDa (for rVP1) and 24.86 kDa (for rVP2) were observed in the insoluble fraction of the bacterial lysate containing the pET-28a-VP1 and pET-28a-VP2 vectors, respectively, while no protein band of similar size was observed in the insoluble fraction of the wild-type lysate (Figure 1C). To purify rVP1 and rVP2, we treated urea to denature the inclusion bodies and performed purification using the Ni-NTA column system. Each recombinant protein was purified effectively after two cycles of washing/elution (Figure 1D).

### 3.2. Reactivity Test of Antiserum by Western Blot and ELISA

Mice injected with the purified rVP1 and rVP2 effectively produced polyclonal antisera pAb-rVP1 and pAb-rVP2, respectively. To determine the reactivity of raised antisera, each antiserum was tested with a serial dilution of rVP1 and rVP2. In Western blot analysis, pAb-rVP1 could detect its antigen (26.73 kDa) at a concentration as low as 500 ng (Figure 2A, lane 3), and pAb-rVP2 could detect its antigen (24.86 kDa) at a loading of as low as 62.5 ng under given experimental conditions (Figure 2B, lane 6). No corresponding protein band was observed when a pre-immune antiserum was used in Western blot analysis (data not shown). To improve the sensitivity of pAb-rVP1 and pAb-rVP2, each polyclonal antiserum, including control mouse serum, was further purified to IgG using a commercial kit. Each IgG was visualized using Coomassie brilliant blue-stained SDS-PAGE gels and used as a primary antibody for Western blot analysis. As expected, pAb-rVP1 and pAb-rVP2 distinctly reacted with respective antigens (Appendix A). To quantitatively measure the reactivity, pAb-rVP1 and pAb-rVP2 were subjected to ELISA using the purified rVP1 and rVP2, respectively. Each pAb reacted positively in ELISA against a series of rVP1 and rVP2 concentrations (500, 250, 125, 62.5, and 31.3 ng/mL). ELISA values of the rVP1 dilution were 0.358, 0.230, 0.163, and 0.114, yielding a regression line (equation: y = 0.0654x − 0.0055) with a correlation coefficient of 0.957 (Figure 2C). ELISA values of the rVP2 dilution were 0.303, 0.229, 0.149, 0.106, and 0.075, yielding a regression line (equation: y = 0.0574x − 0.0007) with a correlation coefficient of 0.981 (Figure 2D).

### 3.3. The Specificity and Efficiency of Polyclonal Antibodies

To test the specificity, pAb-rVP1 and pAb-rVP2 were evaluated with respective homologous and heterologous antigens (rVP1 and rVP2) in the Western blot assay (Figure 3A). After 5 min staining of the membrane with NBT/BCIP, pAb-rVP1 showed clear detection of its homologous antigen rVP1, while pAb-VP2 detected not only the homologous antigen rVP2 but also the heterologous rVP1. After extensive staining (60 min) of the membrane with NBT/BCIP, we found that both pAbs showed a positive reaction with the homologous and heterologous antigens. However, the bands observed for homologous virus antigen–antibody combinations were much stronger than those of heterologous combinations, although each antibody reacted with heterologous antigens in the Western blot. To test the efficiency, pAb-rVP1 and pAb-rVP2 were evaluated with a crude extract from field samples (Figure 3B). Both pAbs showed the corresponding native protein present in the crude extract from AcSBV-infected honeybees (lane 3) and larvae (lane 4) but not in the crude extract from healthy honeybees (lane 2).

### 3.4. Selection of rVP1- and rVP2-Specific mAbs and Their Reactivity to Recombinant Protein and Honeybee Samples

Based on primary screening by Western blot analysis, ten anti-VP1 mAbs in the supernatant of twenty-four clones and ten anti-VP2 mAbs in the supernatant of twenty-four clones were identified as positive clones. Ascites fluids were collected from ascites tumor-forming mice (Appendix A) and further purified to IgG using Proteus Antibody Purification Kit Protein A (Appendix A). Ultimately, positive hybridoma cells secreting mAb-rVP1 and mAb-rVP2 were selected (Figure 4A). Each of these was designated as mAb-rVP1-1 and mAb-rVP2-13 and showed a strongly positive band in Western blot analysis. As expected, mAb-VP1-1 and mAb-VP2-13 bound to only homologous rVP1 (26.7 kDa) and rVP2 (24.9 kDa), respectively, without heterologous antigen binding (Figure 4B). To confirm their specificities, field samples were subject to Western blot analysis using mAb-rVP1-1 and mAb-rVP2-13 (Figure 4C). Each mAb reacted with infected honeybees and larvae as well as the corresponding respective recombinant proteins. Two specific mAbs showed a stronger reaction to AcSBV in larvae than honeybees because the larval stage is the most susceptible to SBV infection. However, they did not react with healthy honeybees (Figure 4C, lane 2). Therefore, each mAb was used for further experiments.

### 3.5. Antigen Detection Limit (Sensitivity) of the IC Strips Using pAb as a Detector Antibody and mAb as a Capture Antibody

The IC assay was designed in the form of a sandwich, using two specific antibodies, mAb-rVP1-1 and mAb-rVP2-13, immobilized on a nitrocellulose membrane at separated test lines, while other specific antibodies, pAb-rVP1 and pAb-rVP2, were conjugated with gold nanoparticles. The gold-bound conjugated pAb-rVP1 and pAb-rVP2 can be bound by a target antigen present in the sample instead of the mAb-rVP1-1 and mAb-rVP2-13 embedded at the test lines. No reaction was observed with the negative control of the IC assay developed in this study (Figure 5). The antibody pair, pAb-rVP1/mAb-VP1-1, showed positive reactivity as low as 1.38 × 10^3^ copies, while the other pair, pAb-rVP2/mAb-VP2-13, showed positive reactivity as low as 1.38 × 10^4^ copies. Therefore, pAb-rVP1/mAb-VP1-1 was selected as a final candidate for the validation.

### 3.6. Validation (Specificity) of the IC Strips Using pAb as a Detector Antibody and mAb as a Capture Antibody

Fifty field samples (various copy numbers from 9.83 × 10^2^ to 1.22 × 10^6^) identified positively by real-time PCR showed a positive reaction in the IC strips coated with antibody pair pAb-rVP1/mAb-VP1-1 (Table 1, Appendix A). On the other hand, 50 samples identified as negative by real-time RT-PCR showed a negative reaction in the IC strips coated with antibody pair pAb-rVP1/mAb-VP1-1 (Table 1, Appendix A). The antibody pair demonstrated high sensitivity and specificity and did not cross-react with the non-infected samples at all. The specificity of mAb-VP1-1 was similar to that of real-time PCR; a different quantity of virus infection was shown. The high agreement between the IC and real-time PCR assays suggests that the IC assay is a very useful method for the rapid and accurate diagnosis of honeybees and larvae infected with AcSBV. A further test with more samples collected nationwide needs to be commercialized as a diagnostic kit for the rapid, sensitive, and specific detection of AcSBV.

## 4. Discussion

SDS-PAGE, Western blot, and ELISA results confirmed that the AcSBV-rVP1 and AcSBV-rVP2 were successfully expressed in *E. coli*. Previously, three structural protein genes (*VP1*, *VP2*, and *VP3*) of CSBV (GenBank accession no. HM237361.1) were successfully expressed in the pGEX-6P-1 bacterial expression system using codon optimization, and each recombinant protein was used for the production of polyclonal antiserum [49]. Based on the sequence alignment, the VP1 and VP2 proteins of AcSBV-Kor share 96.4% and 96.6% amino acid identity with those of CSBV. This suggests that the polyclonal antiserum directed against the recombinant proteins (structural proteins) of AcSBV-Kor can cross-react with their structural protein counterparts in CSBV.

AcSBV-VP2 comprises the N-terminal end (residue 213) to residue 388 of the coat protein, while AcSBV-VP1 comprises residues from 774 to the C-terminal end (residue 965) of the coat protein. AcSBV-VP2 shares 38.4% nucleotide and 69.8% amino acid identity with AcSBV-VP1 based on the sequence alignment. In this study, rVP1 and rVP2 were evaluated for their antigenicity against their specific antibody and each other. In Western blot analysis, pAb-rV1 showed a strong specificity to the homologous antigen and a weak specificity to the heterologous antigen. pAb-rVP2 reacted strongly with both homologous and heterologous antigens. This suggests that pAb-rVP1 is less sensitive but has a higher specificity than pAb-rVP2. Our results also indicate that rVP2 shares major immune-relevant domains with rVP1 and that both recombinant proteins present authentic viral epitopes. On the other hand, mAb-VP1-1 and mAb-VP2-13 were bound only to their homologous antigens. Both pAbs (pAb-rVP1 and pAb-rVP2) and mAbs (mAb-rVP1 and mAb-rVP2) are able to recognize native virus particles in field samples, providing the required basis for the development of the IC kit.

A critical factor of IC assay design is optimal selection of an antibody that shows sensitive and specific detection of a target antigen. Ultimately, the performance of the IC assay relies on the antibody affinity when binding an antigen in a sample. In a typical sandwich format, either a single mAb or two different mAbs are used as a detector antibody and a capture antibody in the IC strips [32]. In the case of low sample loading, the use of the same mAb for detection and capture in the IC assay showed poor detection sensitivity because of competition for the same epitope between the detector and capture mAbs [50]. The use of two different mAbs in the IC assay is an expensive suggestion, limiting its practical usefulness in POCT. Furthermore, the detector mAb binding to an epitope has the possibility of affecting other spatially unrelated epitopes in a way that could modify binding to the capture mAb [51].

Alternatively, mAb has been applied as a detector antibody, while pAb was applied as a capture antibody [32], and vice versa [41]. Utilizing mAb as a detector antibody and pAb as a capture antibody may result in alteration of some of the original epitopes, leaving the rest of the epitopes unaltered and accessible to the polyclonal capture antibodies [32]. Furthermore, pAbs are generally less expensive and faster to produce than mAbs. Hence, the present strip test based on a combination of mAb and pAb provides an acceptable alternative for onsite and cost-effective diagnosis of AcSBV infection.

Antibody pairs that can simultaneously bind to the target antigen with high sensitivity and specificity are required for a sandwich assay. In this study, we developed an IC test strip for detecting AcSBV using two pairs of antibodies (pAb-rVP1/mAb-rVP1-1 and pAb-rVP2/mAb-rVP2-13). This design helps to accumulate the target antigen on the pAb-rVP1- or pAb-rVP2-conjugated pad, therefore enhancing the detection of antigen–antibody binding to increase the sensitivity. As shown in Figure 5, the antibody pair of pAb-rVP1/mAb-rVP1-1 showed more sensitivity than pAb-rVP2/mAb-VP2-13 at detecting low concentrations in a sample. Thus, it was hypothesized that the implications of pAbs, which can detect multiple epitopes simultaneously, may replace at least one mAb in the sandwich format of an IC test [32]. Furthermore, this design helps the antigen–pAb complex to react with mAb-rVP1-1 or mAb-rVP2-13 embedded at the test line, therefore enhancing the capturing specificity. In a preliminary test, an IC kit reacted negatively against deformed-wing virus (DWV)-infected adult bee samples. Thus, it is important to ensure that the capture antibody does not recognize other antigens present in a field sample. Currently, there is no commercially available IC test kit for the detection of AcSBV. The specificities and sensitivity of the newly developed test strips described in the present study are also comparable with the real-time PCR method.

RT-PCR is a simple, sensitive, and cost-effective method to detect SBV RNA based on the amplification of specific viral sequences [9,23,27]. However, this method is limited to quantifying viral RNA and often produces enormous specificity and sensitivity. Real-time RT-PCR is a better tool for sensitive, accurate, and specific detection and quantification of SBV [25,26,29]. Nevertheless, the two methods are time-consuming and require highly trained personnel to handle specialized laboratory instruments. RT-LAMP detected SBV RNA as soon as 20 min at 65 °C, and is suitable for onsite detection in field [24]. Although this method has some advantages, one of the limitations is a high risk of carryover contamination which often produces false-positive results in negative samples [52]. Compared with the traditional detection methods, an IC strip method has the advantages of being easy to use, providing an answer rapidly and at a low cost, and not requiring specialized equipment or technical personnel; thus, such a method is suitable for field testing for antigens.

## 5. Conclusions

This study established a basis for the development of related IC detection kits that are useful for bee farmers and verified their capability for rapidly diagnosis at the time and site of an encounter with environmental samples.

## Figures and Tables

**Figure 1 viruses-13-02439-f001:**
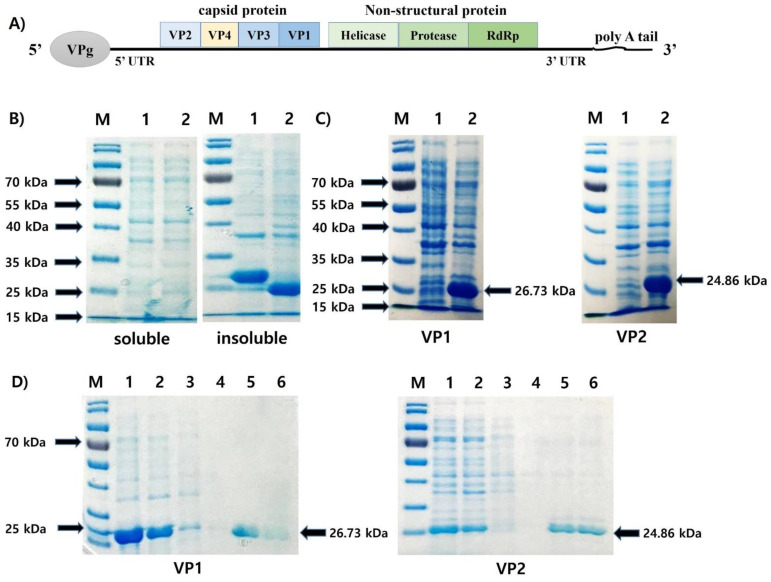
(**A**) Genome organization of AcSBV. (**B**) IPTG (final concentration at 1 mM) induction for the expression of pET-28a vector containing AcSBV-VP1 (lane 1) and AcSBV-VP2 (lane 2). Soluble fraction (left) and insoluble fraction (right) of the *Escherichia coli* BL21(DE3) transformants. (**C**) Insoluble fraction of wild-type (lane 1) and transformant (lane 2) *E. coli* BL21(DE3). (**D**) Purification of the recombinant VP1 (rVP1) and VP2 (rVP2) proteins by Ni-NTA resin chromatography under denaturing conditions. M: protein molecular weight marker; lanes 1–2: cell lysate 1 and 2; lanes 3–4: first washing and second washing; and lanes 5–6: first elution and second elution. Arrows indicate rVP1 (26.73 kDa) or rVP2 (24.86 kDa).

**Figure 2 viruses-13-02439-f002:**
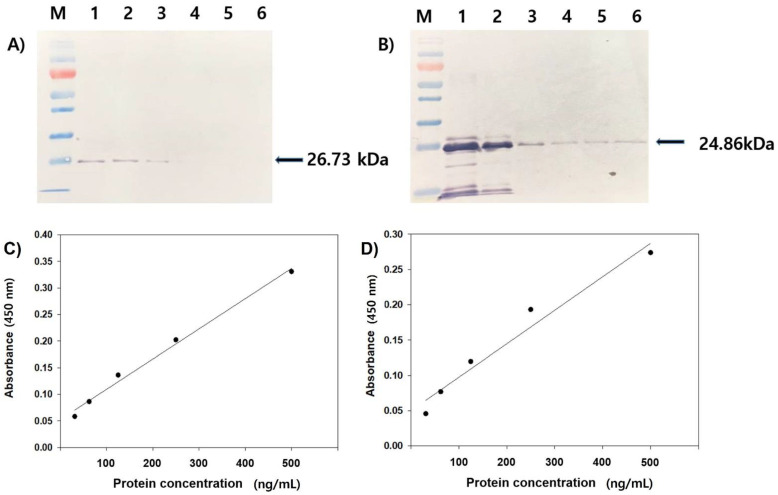
Western blot analysis of the rVP1 (**A**) and rVP2 (**B**) using respective polyclonal antisera (pAb-rVP1 and pAb-rVP2) collected at the sixth week. Lanes 1–5 represent the different concentration (2000, 1000, 500, 250, 125, and 62.5 ng) of each protein. Detection by ELISA of serial dilutions (500, 250, 125, 62.5, and 31.3 ng/mL) of the rVP1 (**C**), and rVP2 (**D**) using the purified IgG (pAb-rVP1 and pAb-rVP2). ELISA values of the rVP1 dilution yielded a regression line (equation: y = 0.0654x − 0.0055, r = 0.957), and the rVP2 dilution yielded a regression line (equation: y = 0.0574x − 0.0007, r = 0.981).

**Figure 3 viruses-13-02439-f003:**
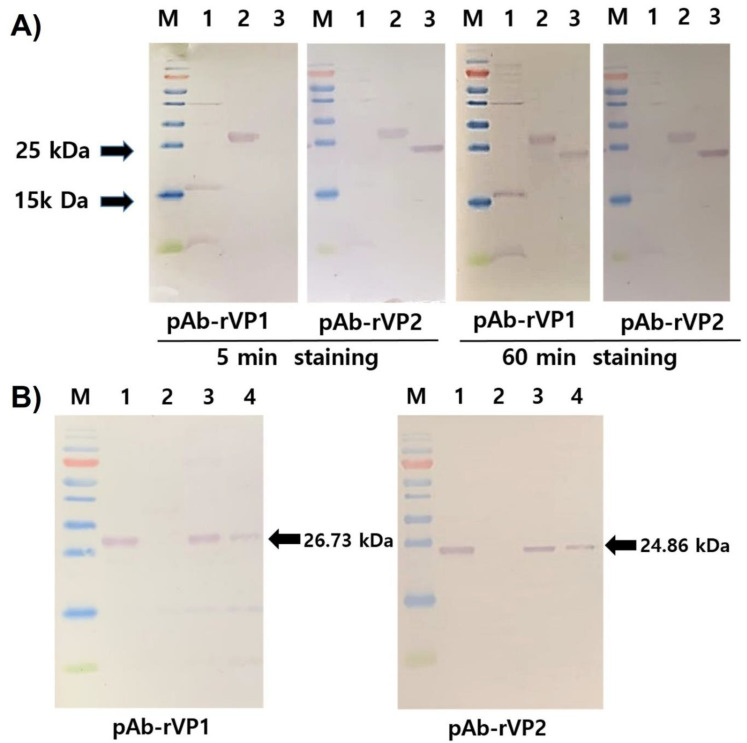
(**A**) Western blot analysis of rVP1 and rVP2 (lanes 1–3, 5 μg each) using pAb-rVP1 and pAb-rVP2 and (**B**) Western blot analysis of rVP1, rVP2 (lane 1, 5 μg each), crude protein extract from healthy honeybees (lane 2, 20 μg), crude protein extract from AcSBV-infected honeybees (lane 3, 20 μg), and AcSBV-infected larvae (lane 4, 20 μg) using pAb-rVP1 (left) and pAb-rVP2 (right).

**Figure 4 viruses-13-02439-f004:**
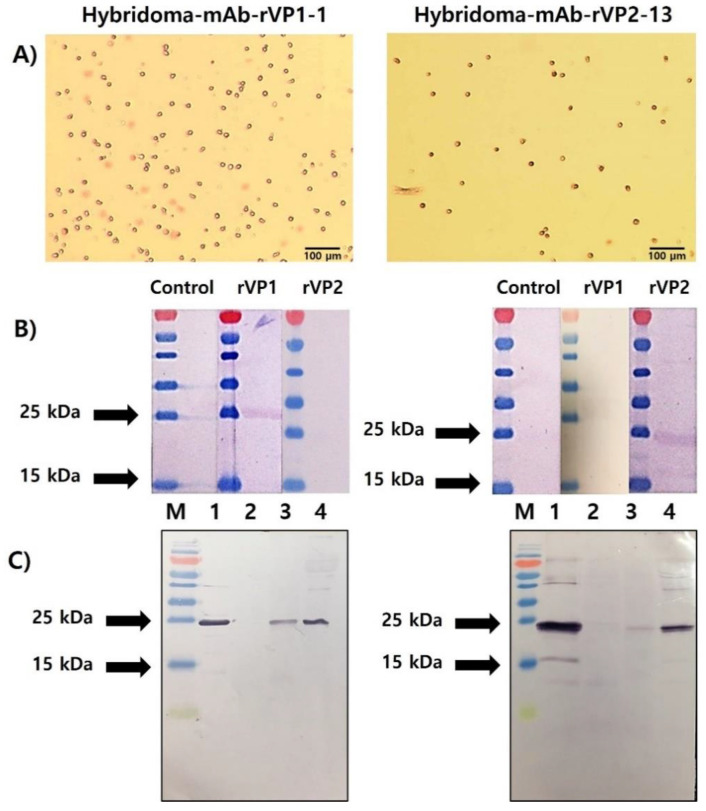
(**A**) Hybridoma clones with the highest titer of anti-rVP1 (left) and anti-rVP2 (right); (**B**) screening by Western blot analysis showed that mAb-VP1-1 and mAb-VP2-13 reacted strongly against rVP1 and rVP2, respectively; and (**C**) immunoreactivity of mAb-VP1-1 and mAb-VP2-13 with field samples in Western blotting. Lane 1: the purified recombinant protein (rVP1, rVP2); lane 2: healthy honeybees, lane 3: infected honeybees; and lane 4: infected larvae.

**Figure 5 viruses-13-02439-f005:**
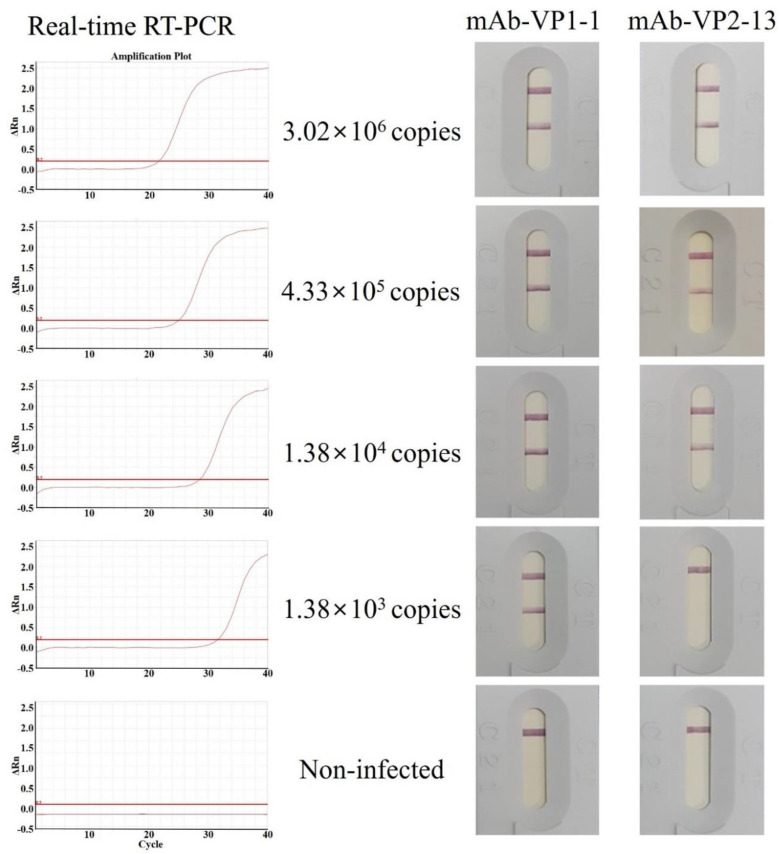
Detection limit of the IC assay with gold-conjugated VP1-specific and VP2-specific mAbs. The reference sample was serially diluted (10-fold) from 10^3^ to 10^6^ copies. Each diluted sample was amplified by real-time RT-qPCR and compared to standard curves in Appendix A.

**Table 1 viruses-13-02439-t001:** Comparative evaluation of AcSBV-infected larva samples by real-time PCR and IC strips coated with pAb-rVP1 as a detector antibody and mAb-VP1-1 as a capture antibody.

Real-Time RT-PCR	Rapid Strip Test		
	Positive	Negative	Total
Positive ^1^	50	0	50
Negative	0	50	50
Total	50	50	100

^1^ copy number: 9.83 × 10^2^~1.22 × 10^6.^

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
