# Peer review of "Development of a Kit for Rapid Immunochromatographic Detection of Sacbrood Virus Infecting Apis cerana (AcSBV) Based on Polyclonal and Monoclonal Antibodies Raised against Recombinant VP1 and VP2 Expressed in Escherichia coli"

_viruses, 2021, doi:10.3390/v13122439_

Round 1

Reviewer 1 Report

11/11/2021

In this manuscript, the authors generated polyclonal and monoclonal antibodies against structural proteins of VP1 and VP2 of a Korean isolate of Sacbrood virus (AcSBV) and developed an IC assay using these antibodies for rapid detection of specific infection of SBV. Overall the developed assay is sensitive and could be used for specific detection of SBV infection.

The manuscript has some typos and needs to be corrected for spelling mistakes.

General comments:

For general readers, a schematic representation showing the organization of the structural proteins (VP2-VP1) as well as non- structural proteins in the Sacbrood virus genome needs to be presented in figure-1A followed by remaining figures.

Specific comments

Abstract:

Line 30: thedevelopment. Needs to correct

Material & Methods:

Line 195.

8000 rpm? Needs to check proper speed.

Line 208-209.

For initial screening after fusion ELISA is preferred to minimize the work load.

Need clarification from authors why they chose Western blotting instead of ELISA to screen antibody. How many clones did they screen? Also, a second round of cloning is needed to ensure clonal purity.

Line: 210

Ascetic fluids: should be Ascites fluids

Generation of Ascites fluids:

For generation of ascites fluids pristine prime to mice is required followed by hybridoma cell injection. What did the authors used for priming?

Line 221-222. Which supplier’s antibody purification kit was used? Bio-Rad or MilliporeSigma?

Results:

Line 303 & 311.

The word “Wild-type” is confusing and needs to replace with control or uninduced.

Line 321: 62.5 ug should be 62.5 ng.

Figure 3: 3B, pAb-rVP3 (right) blot is missing.

Author Response

In this manuscript, the authors generated polyclonal and monoclonal antibodies against structural proteins of VP1 and VP2 of a Korean isolate of Sacbrood virus (AcSBV) and developed an IC assay using these antibodies for rapid detection of specific infection of SBV. Overall the developed assay is sensitive and could be used for specific detection of SBV infection.

The manuscript has some typos and needs to be corrected for spelling mistakes.

 General comments:

For general readers, a schematic representation showing the organization of the structural proteins (VP2-VP1) as well as non- structural proteins in the Sacbrood virus genome needs to be presented in figure-1A followed by remaining figures.

Response: Thank you for your valuable comment and notifying us. We have revised the manuscript according to your comment.

 Specific comments

Abstract:

Line 30: thedevelopment. Needs to correct

Response: We made a mistake and corrected the word in the manuscript. Thank you for your valuable comment and notifying us.

Material & Methods:

Line 195. 8000 rpm? Needs to check proper speed.

Response: Thank you for your valuable comment and notifying us. The speed was 400 × g. I made a mistake during editing the manuscript. We corrected the centrifugation speed in the manuscript.

Line 208-209. For initial screening after fusion ELISA is preferred to minimize the work load. Need clarification from authors why they chose Western blotting instead of ELISA to screen antibody.

Response: As you suggested, ELISA should be better to screen hybridoma cells. The first author was concerned with the colorimetric reaction to develop the IC kit. In this respect, she performed mAbs reaction by Western blot and selected mAbs showing strong reaction with antigens.

How many clones did they screen?

Response: Ten anti-VP1 mAbs in the supernatant of twenty-four clones and ten anti-VP2 mAbs in the supernatant of twenty-four clones were identified as positive clones. We clarified the sentence in lines 379-381.

Also, a second round of cloning is needed to ensure clonal purity.

Response: Thank you for your valuable comment and notifying us. As you mentioned, we performed subcloning the hybridoma cells to achieve monoclonality. During the writing of the manuscript, we didn’t describe a method in detail. We added some sentences about the process (lines 363-365).

Line: 210. Ascetic fluids: should be Ascites fluids

Response: We made a mistake and corrected the word in the manuscript. Thank you for your valuable comment and notifying us.

Generation of Ascites fluids: For generation of ascites fluids pristine prime to mice is required followed by hybridoma cell injection. What did the authors used for priming?

Response: Thank you for your valuable comment and notifying us. The stimulants were administered in amounts of 0.5 ml of Freund's incomplete adjuvant per a mouse. We revised the sentence in lines 194-195 and 500 in the manuscript as well as Figure S2 in Supplementary Materials.

Line 221-222. Which supplier’s antibody purification kit was used? Bio-Rad or MilliporeSigma?

 Response: MilliporeSigma is correct. We described the manufacturer in the text (line 136).

Results:

Line 303 & 311.

The word “Wild-type” is confusing and needs to replace with control or uninduced.

Response: Thank you for your valuable comment and notifying us. We have revised the manuscript according to your comment.

Line 321: 62.5 ug should be 62.5 ng.

Response: Yes, you’re right. Sorry for the mistake and we re-wrote the word in the manuscript. Thank you for your valuable comment and notifying us.

Figure 3: 3B, pAb-rVP3 (right) blot is missing.

Response: We made a mistake and corrected the word in the manuscript. Thank you for your valuable comment and notifying us.

Reviewer 2 Report

Sacbrood virus is one of the most destructive honeybee viruses that causing bee losses in Apis cerana all over the Asian apiculture. In current manuscript, authors developed a kit for rapid detection of sacbrood virus of A. cerena based on polyclonal and monoclonal antibodies against recombinant VPI and VP2, two of the sacbrood virus capsid proteins. It seemed that the antibody was successfully prepared by confirming the sentitivity and validation of the IC strips. The currrent developed kit might be employed in application for detecting the pathogen of sacbrood virus and protecting the Asian honeybee population in practice in future.

For better improve the quality of the manuscirpt, the following questions should be adressed.

1.In materials and methods. Some protocols might be described concisely if they have been well discribed in other papers. e.g in 2.2, and in 2.5.

2.In current manuscript, authors identified the various copy numbers by real time PCR (Line 32, Line 33, Line 292, Line 392, et al) and concluded that the “positive reactivity as low as 1.38×103 copies” (in abstract part Line 32) and “positive reactivity as low as 1.38×104 copies” (Line 33). However, in present manuscript, the authors have not demonstrated the “copy” in this manuscript refered to the “copy of virus” or the others? And how the copies were caculated?

3.The infected honeybees by sacbrood virus normally coinfected by other viruses, no matter the bee is in larva or in adult. To better evaluate the specificity of the detecting kit, whether the kit has been tested to detect other honeybee viruses, such as deformed wing virus, or black queen cell virus?

4.In discussion. There have several detecting methods related to sacbrood virus have been developed up to date, suggest authors summerized and discussed them accordingly.

5.Some other minor mistakes should be corrected. Such as:

(1) in Line 30, “thedevelopment” should be corrected as: the development.

(2) in figure 4, B, mistake in format of bracket.

(3) In figure 4, “Hybridoma clones with the highest titer of anti-rVP1 (left) and -rVP2 (right)”, the authors missed to note the magnification times in figure legends and missed the bar of scale in figure.?

(4) In figure 5 and supplementary figures, all the figures according to Real-time RT-PCR displayed with low quality. The numbers in x-axis and the y-axis can not been clearly viewed.

Author Response

Sacbrood virus is one of the most destructive honeybee viruses that causing bee losses in Apis cerana all over the Asian apiculture. In current manuscript, authors developed a kit for rapid detection of sacbrood virus of A. cerena based on polyclonal and monoclonal antibodies against recombinant VPI and VP2, two of the sacbrood virus capsid proteins. It seemed that the antibody was successfully prepared by confirming the sensitivity and validation of the IC strips. The current developed kit might be employed in application for detecting the pathogen of sacbrood virus and protecting the Asian honeybee population in practice in future.

For better improve the quality of the manuscript, the following questions should be addressed.

1.In materials and methods. Some protocols might be described concisely if they have been well described in other papers. e.g in 2.2, and in 2.5.

Response: Based on your suggestion, we briefly described the methods in subsections 2.2, 2.3, 2.4, 2.5, 2.6, and 2.7.

2.In current manuscript, authors identified the various copy numbers by real time PCR (Line 32, Line 33, Line 292, Line 392, et al) and concluded that the “positive reactivity as low as 1.38×103 copies” (in abstract part Line 32) and “positive reactivity as low as 1.38×104 copies” (Line 33). However, in present manuscript, the authors have not demonstrated the “copy” in this manuscript referred to the “copy of virus” or the others? And how the copies were calculated?

Response: Thank you for your valuable comment and notifying us.

For the standard of DNA copy number, we used pET28a-VP1 including a 579 bp insert and pET28a-VP2 including a 528 bp insert. For real-time PCR, a primer set was used to amplify a part of the VP1 gene (forward: 5’-CATGGAGAGACAAA GGCGATAC-3’ and reverse: 5’-GCTTCTACCCACACAGACATATT-3’) and the VP2 gene (forward: 5’-TGCGCGCCCA ATACTATAC-3’ and reverse: 5’-CTTTCCCGCACTGAAACTTATTAC-3’). A ten-fold serial dilution series of the pET28a-VP1 and pET28a-VP2 ranging from 1 × 103 to 1 × 106 copies/μL, was used to construct the standard curves for both VP1 andVP2.

The concentration of the plasmid was measured using a fluorometer and the corresponding copy number was calculated using the following equation: DNA (copy) = 6.02 × 1023 (copy/mol) × DNA amount (g) / DNA length (bp) × 660 (g/mol/bp) (Lee et al., 2006, J. Biotechnol. 123, 273-280). The plot of a standard curve of Ct values against the logarithmic dilutions produced an R2 value of 0.9965 and a regression line (equation: y = -3.8356x + 42.7930) for the VP1 and an R2 value of 0.9933 and a regression line (equation: y = -3.7314x + 43.5159) for the VP2. The data was shown as Figure S4 in Supplementary Materials.

To clarify your suggestion, we have added some sentences above in lines 271-280 and 504-509 in text and Figure S4 of Supplementary Materials.

3.The infected honeybees by sacbrood virus normally coinfected by other viruses, no matter the bee is in larva or in adult. To better evaluate the specificity of the detecting kit, whether the kit has been tested to detect other honeybee viruses, such as deformed wing virus, or black queen cell virus?

Response: Thank you for your good suggestion. Both Deformed wing virus (DWV) and Black queen cell virus (BQCV) have been reported from Apis mellifera adults and larvae in Korea. We collected Apis mellifera adult honeybees in the apiary and detected the presence of DWV infection by real-time PCR using virus-specific primers (DWV-UF-F3: 5’-GTTGTTTGAGAACCCAACTTG GTTGTTTGAGAACCCAACTTG-3’ and DWV-UF-R3: CGCTTGCAACCACACTTTCA: Journal of Apiculture 28:237-244) amplifying 133 bp. Five samples were tested for the presence of DWV with the IC kit developed, showing a negative reaction. Unfortunately, we could not test BQCV due to the absence of samples. We will collect BQCV-infected samples in the next spring season, 2022, and test any reaction with the IC kit.

To clarify your suggestion, we have added the sentence in lines 469-470 as follows: In a preliminary test, an IC kit reacted negatively against Deformed wing virus-infected adult bee samples.

4.In discussion. There have several detecting methods related to sacbrood virus have been developed up to date, suggest authors summarized and discussed them accordingly.

Based on your suggestion, we accordingly discussed the conventional methods in lines 475-484.

5.Some other minor mistakes should be corrected. Such as:

(1) in Line 30, “the development” should be corrected as: the development.

Response: We made a mistake and corrected the word in the manuscript. Thank you for your valuable comment and notifying us.

(2) in figure 4, B, mistake in format of bracket.

Response: We revised Figure 4. Thank you for your valuable comment and notifying us.

(3) In figure 4, “Hybridoma clones with the highest titer of anti-rVP1 (left) and -rVP2 (right)”, the authors missed to note the magnification times in figure legends and missed the bar of scale in figure.?

Response: We revised Figure 4. Thank you for your valuable comment and notifying us.

(4) In figure 5 and supplementary figures, all the figures according to Real-time RT-PCR displayed with low quality. The numbers in x-axis and the y-axis can not been clearly viewed.

Response: We revised Figure 5 and Supplementary Figures. Thank you for your valuable comment and notifying us.